# Unselected Population Genetic Testing for Personalised Ovarian Cancer Risk Prediction: A Qualitative Study Using Semi-Structured Interviews

**DOI:** 10.3390/diagnostics12051028

**Published:** 2022-04-19

**Authors:** Faiza Gaba, Samuel Oxley, Xinting Liu, Xin Yang, Dhivya Chandrasekaran, Jatinderpal Kalsi, Antonis Antoniou, Lucy Side, Saskia Sanderson, Jo Waller, Munaza Ahmed, Andrew Wallace, Yvonne Wallis, Usha Menon, Ian Jacobs, Rosa Legood, Dalya Marks, Ranjit Manchanda

**Affiliations:** 1Wolfson Institute of Population Health, Barts CRUK Centre, Queen Mary University of London, Old Anatomy Building, Charterhouse Square, London EC1M 6BQ, UK; f.gaba@qmul.ac.uk (F.G.); s.oxley@qmul.ac.uk (S.O.); xintingliu@yahoo.co.uk (X.L.); d.chandrasekaran@qmul.ac.uk (D.C.); 2Department of Gynaecological Oncology, St Bartholomew’s Hospital, London EC1A 7BE, UK; 3Strangeways Research Laboratory, Centre for Cancer Genetic Epidemiology, Department of Public Health and Primary Care, The University of Cambridge, Cambridge CB1 8RN, UK; xy249@medschl.cam.ac.uk (X.Y.); aca20@medschl.cam.ac.uk (A.A.); 4Department of Women’s Cancer, University College London, Gower St, Bloomsbury, London WC1E 6BT, UK; j.k.kalsi@ucl.ac.uk; 5Department of Clinical Genetics, University Hospital Southampton NHS Foundation Trust, Tremona Rd, Southampton SO16 6YD, UK; lucy.side@uhs.nhs.uk; 6Early Disease Detection Research Project UK (EDDRP UK), 2 Redman Place, London E20 1JQ, UK; saskia.sanderson@ucl.ac.uk; 7Cancer Prevention Group, King’s College London, Great Maze Pond, London SE1 9RT, UK; jo.waller@kcl.ac.uk; 8North East Thames Regional Genetics Unit, Department Clinical Genetics, Great Ormond Street Hospital, London WC1N 3JH, UK; munaza.ahmed@gosh.nhs.uk; 9Manchester Centre for Genomic Medicine, 6th Floor Saint Marys Hospital, Oxford Rd, Manchester M13 9WL, UK; andrew.wallace@mft.nhs.uk; 10West Midlands Regional Genetics Laboratory, Birmingham Women’s NHS Foundation Trust, Birmingham B15 2TG, UK; y.wallis@nhs.net; 11Medical Research Council Clinical Trials Unit at UCL, Institute of Clinical Trials and Methodology, University College London, 90 High Holborn, London WC1V 6LJ, UK; u.menon@ucl.ac.uk; 12Department of Women’s Health, University of New South Wales, Sydney 2052, Australia; i.jacobs@unsw.edu.au; 13Faculty of Public Health and Policy, London School of Hygiene & Tropical Medicine, 15-17 Tavistock Place, London WC1H 9SH, UK; rosa.legood@lshtm.ac.uk (R.L.); dalya.marks@lshtm.ac.uk (D.M.); 14Department of Gynaecology, All India Institute of Medical Sciences, New Delhi 110029, India

**Keywords:** ovarian cancer, population testing, risk stratification, health and well-being

## Abstract

Unselected population-based personalised ovarian cancer (OC) risk assessments combining genetic, epidemiological and hormonal data have not previously been undertaken. We aimed to understand the attitudes, experiences and impact on the emotional well-being of women from the general population who underwent unselected population genetic testing (PGT) for personalised OC risk prediction and who received low-risk (<5% lifetime risk) results. This qualitative study was set within recruitment to a pilot PGT study using an OC risk tool and telephone helpline. OC-unaffected women ≥ 18 years and with no prior OC gene testing were ascertained through primary care in London. In-depth, semi-structured and 1:1 interviews were conducted until informational saturation was reached following nine interviews. Six interconnected themes emerged: health beliefs; decision making; factors influencing acceptability; effect on well-being; results communication; satisfaction. Satisfaction with testing was high and none expressed regret. All felt the telephone helpline was helpful and should remain optional. Delivery of low-risk results reduced anxiety. However, care must be taken to emphasise that low risk does not equal no risk. The main facilitators were ease of testing, learning about children’s risk and a desire to prevent disease. Barriers included change in family dynamics, insurance, stigmatisation and personality traits associated with stress/worry. PGT for personalised OC risk prediction in women in the general population had high acceptability/satisfaction and reduced anxiety in low-risk individuals. Facilitators/barriers observed were similar to those reported with genetic testing from high-risk cancer clinics and unselected PGT in the Jewish population.

## 1. Introduction

In recent years, multidisciplinary management of ovarian cancer (OC) has led to improved care and cancer outcomes for patients [1,2,3]. Nevertheless, OC causes significant mortality due to the fact of its aggressive nature, late stage at presentation [4] and lack of an effective screening programme for women in the general population [5]. Therefore, effective and targeted preventive strategies are necessary, and their validation and implementation require similar collaboration among disciplines. Currently, OC and breast cancer (BC) prevention is targeted at high-risk individuals carrying mutations in cancer-susceptibility genes (CSGs), identified through clinical criteria/family history (FH)-based testing. A prime example are the *BRCA1/BRCA2* carriers with a 17–44% OC risk and 69–72% BC risk until age 80 years [6]. More recently, moderate-penetrance OC CSGs (*RAD51C*: lifetime OC risk = 11% [7]; *RAD51D*: lifetime OC risk = 13% [7]; *BRIP1*: lifetime OC risk = 5.8% [8]) and ~30 validated single-nucleotide polymorphisms (SNPs), which can be combined into a polygenic risk score, were validated [9,10]. High-risk women can opt for clinically effective interventions of screening/prevention to reduce cancer burden. However, the current clinical criteria/FH-based genetic testing approach misses >50% of mutation carriers [11,12,13], is associated with limited uptake and underutilisation of genetic testing and requires women to get cancer before identifying unaffected individuals to prevent cancer, all of which results in delayed carrier identification [14,15]. For example, 97% of estimated *BRCA* carriers remain unidentified [15]. This reflects huge missed opportunities for precision prevention.

A “precision prevention” approach incorporates individual variability into various risk factors, whether genetic or non-genetic (e.g., environmental, hormonal, lifestyle and behavioural). This encompasses preventing disease occurrence (i.e., primary prevention) as well as screening/early detection of pre-symptomatic disease (i.e., secondary prevention). OC risk prediction models incorporating validated SNPs (polygenic risk score) together with moderate–high penetrance genetic and epidemiologic/hormonal data are now available [16]. This improves precision of OC risk estimation, allowing for population division into risk strata for targeted downstream risk-stratified screening and/or prevention for those at increased risk [9,17]. We have shown that it is cost effective to undertake OC surgical prevention at a 4–5% lifetime OC risk [18,19].

Unselected population genetic testing (PGT) can overcome the limitations of current clinical testing. PGT identifies many more at-risk mutation carriers through the use of validated risk models, incorporating non-genetic and genetic factors for predicting personalised lifetime OC risk. This allows for OC risk stratification for targeted precision prevention. The current evidence for PGT largely comes from the Ashkenazi Jewish (AJ) population. Data from UK, Canadian and Israeli studies [11,12,13] show that AJ population-based *BRCA* testing is acceptable, feasible and deliverable in a community setting; identifies 100% of additional carriers; has high satisfaction rates (i.e., 90–95%); does not harm psychological health/quality of life (QoL) [11,12,20]; reduces long-term anxiety [21]; is extremely cost effective (cost savings in most scenarios) for UK/US health systems [22,23]. In general population surveys, we found 75% acceptability towards population testing for CSGs for OC risk stratification with 72% signalling adoption of positive health behaviours following OC risk disclosure [24,25]. We also showed that unselected PGT for a BC/OC gene panel (i.e., *BRCA1/BRCA2/RAD51C/RAD51D/BRIP1/PALB2*) was more cost effective than current clinical criteria/FH-based testing approaches and can prevent thousands of OC/BC cases [26].

Unselected PGT and OC risk stratification through personalised OC risk prediction has been investigated as part of a pilot precision prevention cohort study (i.e., PROMISE Feasibility Study, ISRCTN54246466) [17]. Quantitative data from our group suggest population-based personalised OC risk stratification is feasible and acceptable, has high-satisfaction, reduces cancer-worry/risk perception and, overall, does not negatively affect psychological well-being/quality of life [17]. However, prospective qualitative data from the general population are lacking. Policymakers and commissioners require qualitative data from service users to inform the development of guidelines and care pathways, as this provides in-depth understanding of the reasons for the choices made. In this paper, we report on the attitudes, experiences, emotional well-being and health of women who received low-risk results following unselected PGT/OC risk stratification for OC precision prevention.

## 2. Materials and Methods

### 2.1. PROMISE Feasibility Study Design

A qualitative study was set within a prospective cohort, pilot/feasibility study (PROMISE FS; ISRCTN:54246466) that evaluated stratification of a general population using a validated risk prediction model incorporating genetic and non-genetic risk factors for personalised, predicted lifetime OC risk and offered downstream risk-management options for screening/prevention [17]. One hundred and three unselected women, from North East London, ≥18 years of age and with no personal history of OC or prior ovarian CSG testing, were recruited via primary care. They received pre-test written information, access to a mandatory online bespoke decision aid and an optional telephone helpline. Participants provided a blood sample for PGT (i.e., *BRCA1/BRCA2/RAD51C/RAD51D/BRIP1* and 30 OC SNPs). A risk prediction algorithm incorporating genetic and epidemiological data provided a personalised predicted lifetime OC risk. Participants were stratified into risk categories (i.e., low risk = <5%; intermediate risk = ≥5%–<10%; high risk = ≥10%) and offered downstream management accordingly (low risk = lifestyle advice; intermediate/high risk = lifestyle advice, ROCA (risk of ovarian cancer algorithm)-based OC screening and risk-reducing surgery).

### 2.2. Qualitative Study Design

This nested qualitative study aimed to evaluate the attitudes, experiences, emotional-well-being and health of women following unselected PGT/OC risk stratification for OC prevention. Following test results, interviewees were selected based on sociodemographics (i.e., age, ethnicity, marital status and employment) to ensure representation of the PROMISE cohort. A grounded theory framework was used to construct theory from data, systematically obtained and analysed using an inductive and comparative analysis [27]. In-depth semi-structured one-to-one interviews were conducted, either face to face or via telephone, depending on the participant’s preference, using a predeveloped topic-guide (Appendix A). Topic guide development was informed by a literature review and expert consultation. The wording and sequencing of questions were left open with probes used to elicit more information where appropriate. The interviewer (F.G.) is an academic and gynaecologist by background with training and previous experience in conducting qualitative interviews, and they had no relationship to the participants. A pilot interview was conducted to ensure that it was feasible to cover the entire contents of the topic guide in a single interview setting without participant fatigue and to refine questions. Questions covered: background (i.e., family composition, support network, occupation and hobbies); health values; previous PGT knowledge; reasons for undergoing PGT/OC risk stratification; effect of results on physical and psychological health; advantages and disadvantages of PGT/risk stratification; pre-test information; satisfaction and regret. Following informed consent, interviews were audio-recorded and transcribed verbatim. Interviews were conducted until informational saturation was reached [28].

The study was approved by the London–Central Research Ethics Committee (16/LO/2075) and funded by Cancer Research UK and The Eve Appeal.

### 2.3. Data Analysis

Data were managed using NVIVO version 12 software. The aim of the analysis was to directly reflect the views and experiences of participants and not those predetermined by researchers. Two researchers (F.G. and X.L.) independently coded all transcripts, following a three-step process: open coding, axial coding and selective coding. First, all meaningful aspects of text were labelled starting with a line-by-line analysis (open coding). Second, open codes were categorised, grouping similar codes and refining and combining them into larger themes (axial coding). Through multiple, iterative discussions, we reflected on potential relationships between codes and continued to develop an in-depth understanding of the themes. Coding disagreements were resolved through discussion and further review of the transcripts to reach a consensus. Third, transcripts were reviewed again to ensure themes reflected the data and important ideas and views had not been missed or overrepresented. Selective coding involved integration and refinement of the themes [29].

## 3. Results

### 3.1. Participant Characteristics

Among the PROMISE FS cohort, 102/103 participants received low-risk results, 0/103 intermediate risk, and 1/103 high-risk, and it was therefore decided to focus recruitment on those with low-risk results to enable information saturation.

Nine of one hundred and two low-risk (<5% lifetime risk) individuals who underwent unselected PGT and OC risk stratification within the PROMISE FS were interviewed to achieve informational saturation. There were no interview refusals. Participant characteristics are summarised in Table 1. Ages ranged from 33 to 85 years and the interviews occurred 95–120 days after receipt of the test results and lasted 40–65 min (mean = 53 min). No participant fulfilled the standard NHS clinical criteria for genetic testing.

### 3.2. Themes

#### 3.2.1. Health Beliefs

ID-01: *“… health is priority over everything because if I don’t have my health, there is no point me working or anything else…”*

All stated health was important and described various lifestyle choices in which they engaged. Five exercised regularly, all were currently non-smokers and either teetotallers or consciously limited alcohol intake. Four took steps to improve eating habits by consuming more fruit/vegetables and less red/processed meat. When asked about effects of social media on health-related choices, seven women felt it did not influence lifestyle choices and perceived it as a negative influence, as it distorted the notion of a normal body image and was an unreliable/biased source of information. Two others (ID-05,08) felt that their health choices were positively influenced by social media, motivating them to maintain an active lifestyle and that it was a useful source of information.

Motivators for maintaining good health were having children, personal experience of ill health, witnessing relatives or friends experience ill health, having a physically demanding job, wanting to maintain independence, encouragement from friends or family, sense of duty to self and a desire to feel good. Motivation was rooted in the desire to continue to be functioning members of society.

#### 3.2.2. Decision Making

ID-04: *“I think I’d probably already decided that I was keen to do it but obviously it’s useful to have the full information…”*

Seven women indicated they had decided to undergo PGT/OC risk stratification before receiving pre-test information. However, they felt the information received provided them with additional facts, including advantages/disadvantages, and helped emphasise the gravity and implications of PGT/risk stratification.

ID-05: *“…the questions were important because it made you think about certain things I suppose, which you might not have necessarily thought about”*

When asked about the online decision aid, women reported that it was easy to access and use and provided them with the opportunity to consider the emotional impact of PGT/risk stratification. The eldest woman interviewed (85 years) felt the online component may put off some older women from PGT who lack the technological skills or equipment to complete it.

ID-09: *“It was reassuring that there was a real person I could talk to if I needed to”*

Women found the optional telephone helpline useful, as it enabled them to ask questions and was a source of support for those who needed it. All thought that speaking to a clinician/health professional should be optional and not compulsory. If made compulsory, it may act as a deterrent to people who would perhaps find the process too prescriptive.

ID-08: *“I tend to absorb things better in writing and then you can go back and read over it again”*

The written literature was found to be informative, easy to understand and a good memory aid that the women kept for future reference.

ID-07: *“It probably would have been more difficult to find out that you were high risk but I never thought I would be, because of my knowledge of my own health…”*

Four women considered the implications of being found to be at increased OC risk before deciding to undergo PGT/risk stratification. Five discussed their decision with relatives or friends prior to consent, and they said that their reactions were positive. However, one participant (ID-03) stated that if the reaction of her relatives had been negative, it may have resulted in her declining testing.

When asked if they would have made the same decision if they were older, eight said that they would still have undergone PGT/risk stratification. One participant, aged 69 years (ID-06), questioned whether the benefit of learning her OC risk would be as beneficial if she was older with less life to live but would, nevertheless, still be tested. However, one participant, aged 85 years (ID-09), felt that her decision to undergo PGT would depend on whether she was mobile enough to attend an appointment for her blood test.

Seven felt that they would still have undergone PGT/risk stratification if they were younger. However, two would have declined, either due to the fact of less worry about developing OC (ID-02) or because being found to be a carrier might have deterred her from having children despite the option of in vitro fertilisation (IVF) and preimplantation genetic diagnosis (PGD), which she objected to on religious grounds (ID-01).

#### 3.2.3. Influencing Factors Determining Acceptability

Table 2 summarises the 17 facilitators and 5 barriers identified to influence acceptability and uptake of unselected PGT/OC risk stratification. Facilitators/barriers were categorised as social, demographic, psychological and logistical. Table 2 provides respective quotes and in-depth explanations for each facilitator/barrier.

#### 3.2.4. Effect of Results on Health and Well-Being

ID-03: *“It was a relief”*

All women felt a sense of relief after receiving their <5% predicted lifetime risk (low risk) result. There was no change in alcohol intake, smoking habits, diet or physical activity in any of these individuals.

ID-06: *“They’ve said it’s low-risk and then one day in 10-years’ time, I could wake up and have all the signs of it [ovarian cancer] but ignore it and then it could be a problem”*

One woman felt that being told she was at low risk for OC may lead to a false sense of security and result in her ignoring potential future OC symptoms.

#### 3.2.5. Results Communication

Six individuals shared their results with relatives/friends. The main reason given for sharing results with relatives was because *“they have a right to know as it could also affect them as they are genetically linked”* (ID-01).

#### 3.2.6. Satisfaction

All participants felt that if the situation presented itself again, they would again choose to undergo PGT/OC risk stratification.

ID-02: *“I guess if the worst thing were to happen and I was to develop ovarian-cancer, then I may think that test told me that I was so unlikely to get it and now here I am with it…”*

Feelings of regret were rooted in the notion of having a false sense of security provided by the test and risk stratification. Three felt they might regret being tested if they developed OC in the future.

ID-03: *“I’d say it can potentially save your life”*

All respondents would advise others to get tested, because it would enable identification of mutation carriers or those at increased risk who otherwise would have gone undetected/unidentified. This would enable the individual to access risk-reducing options and, for some, potentially benefit relatives through predictive testing. One woman (ID-01) indicated she would advise women desiring future fertility to consider (before PGT) the implications of passing down a mutation to their child if found to be a carrier. Another participant’s (ID-03) advice was to consider the emotional implications along with coping mechanisms if the individual was found to be a carrier prior to agreeing to PGT/risk stratification.

## 4. Discussion

Our data identified six interconnected themes that affected PGT and personalised OC risk prediction. All interviewees who underwent PGT/OC risk stratification valued health and reported high levels of satisfaction with their decision, reduced anxiety following receipt of negative/low-risk results and would recommend testing to others. All found the helpline facility useful and preferred this to be optional and not compulsory. Seven individuals decided to undergo PGT/OC risk stratification prior to receiving pre-test information. Prior to testing, not all individuals had considered the possibility of a high-risk result, and half had discussed PGT with relatives/friends. There was the feeling that a low-risk category may create a false sense of security leading to individuals potentially ignoring future OC symptoms. The main facilitators/barriers identified fell into demographic, psychological, social and logistical domains (Table 2).

This is the first study to generate qualitative data on unselected PGT and OC risk stratification in women in the general population. We followed a robust qualitative methodology using semi-structured, in-depth interviews until data saturation was reached, and two researchers independently analysed the data and followed a comprehensive three-step coding process. Our sample of women with low-risk results included women of different ethnicities and ages. Limitations include a lack of interviewees declining PGT/risk stratification and a lack of participants who were intermediate/high risk. Further interviews with this sub-group of women are necessary to explore any differences in the impact of unselected PGT/OC risk stratification on health and well-being in women who were identified as being at increased risk and carrying a high penetrant mutation. In addition, all interviews were conducted three–four months following receipt of the results. We did not explore the longer-term implications on health/well-being.

A few OC risk prediction models have been published previously. However, these have been restricted to largely epidemiological/family-history-based factors [31,32]. Pearce et al. incorporated an SNP-based genetic score along with an epidemiological/hormonal/family-history-based assessment, but this model had fewer SNPs and did not include moderate–high penetrance cancer susceptibility genes [33]. The risk prediction algorithm used in our study was more comprehensive, as it incorporates both genetic factors, in terms of established cancer susceptibility genes, as well common genetic variants (polygenic risk score) along with non-genetic (i.e., epidemiological, family history, hormonal and reproductive) factors in women in the general population [16]. This has been validated in a prospective data set from the UKCTOCS general population screening trial [5,16].

Qualitative data from this study corroborate earlier published quantitative analysis from the PROMISE study, confirming the feasibility and acceptability of undertaking population testing for personalised OC risk prediction [17]. We demonstrated acceptability and high satisfaction with delivering PGT and OC risk stratification without formal pre-test counselling in a general population. A web-based direct-to-patient model provides an attractive scalable option for PGT, and it was also found to be effective in an Australian AJ population study [34]. High satisfaction without formal pre-test counselling was previously reported in Australian [34], Israeli [35] and Canadian [20] Jewish population testing studies. The finding that all participants valued health, were highly motivated and actively engaging with healthy lifestyle choices was similar to several reports from general population individuals undergoing direct-to-consumer (DTC) genome-wide profiling for chronic disease risk [36]. A US study investigating psychological/behavioural impact of DTC genome-wide profiling found that 88% self-reported his/her health to be either “good” or “very good” [36]. Most participants were already meeting/exceeding healthy lifestyle guidelines, and no significant differences between baseline and follow up in dietary fat intake or exercise behaviour were observed [36]. Thus, individuals undergoing unselected DTC genomic testing appear to represent a population already engaging in healthy behaviours. Similarly, in our self-selected study population, we found no change in lifestyle habits following delivery of results in low OC risk women. Our finding of no lifestyle change following results disclosure is contrary to those by Meisel [25], who investigated anticipated health behaviour following disclosure of genetic risk for breast/ovarian cancer and found 72% of 837 women at population-level risk indicated they would pursue a healthy lifestyle. This difference may be due to the qualitative study design and intention–action gap, where people’s intention to make lifestyle changes exceeds actual behavioural change. Similarly to Meisel, we found that genetic testing empowered individuals [25].

Unlike a DTC study [36] that showed no difference in anxiety, we found reduced anxiety post results in these low-risk individuals. Similar results of reduced anxiety, overall, were reported in Jewish population testing studies [11,21], although with increased short-term anxiety in mutation carriers [20,35,37]. More qualitative data are required on the short- and long-term effects of intermediate–high risk results on individuals undergoing unselected PGT.

The finding that the majority of interviewees had decided to undergo OC risk stratification even before receiving the pre-test information was similar to findings in Jewish population-based *BRCA* testing, where we found a 60% “intention to test” at the outset and an 88% final uptake [38]. Inherent lay beliefs and the views and experiences of genetic testing of relatives and friends may influence the decision-making process. A UK qualitative study on motivations and attitudes towards predictive *BRCA* testing reported that interviewees had decided to undergo testing before pre-test counselling, based on information provided by probands and not information received during counselling [39]. Similarly, qualitative US [40] and Italian [41] studies evaluating pre-test genetic counselling for *BRCA1/BRCA2/MLH1* revealed that approximately 50% of women had decided to undergo genetic testing before counselling and irrespective of practitioner/counsellor advice. Pre-test information received during counselling is important and beneficial, but it may not impact on the final decision for many [41].

Our results highlight that decision making and acceptability are a dynamic process that changes over time. Although, the majority of interviewees reported they would have still undergone testing if they were younger/older, two stated they would have declined if they were younger and one if they were older. The reasons for declining when younger included determent from having children if CSGs were identified and health being less of a priority when young. These finding are consistent with the literature investigating the experiences of individuals undergoing predictive *BRCA1/BRCA2* testing [39]. Reduced mobility and potential lack of technology/internet skills amongst older participants highlighted as impediments reflect the need to develop flexible community-based or direct-to-consumer approaches for unselected PGT/OC risk stratification if it is to be equally accessible to all age groups.

Facilitators (Table 2) for PGT identified by us have also been widely reported elsewhere amongst individuals undergoing predictive, clinical criteria or unselected genetic testing in the AJ population [38,39,40,42,43,44,45,46,47]. Additional facilitators identified from our data include the existence of preventative OC measures, future regret if the individual developed OC, simplicity and ease of testing, difficulty in OC early detection and previous involvement in OC research studies. Interestingly, unknown FH was identified as a facilitator, whereas aa lack of FH was identified as a barrier amongst individuals undergoing criteria-based genetic testing as well as amongst clinicians [40,48,49]. The barriers we identified to PGT have also been previously reported for individuals undergoing clinical testing [49,50]. No known prevention for a condition that is being tested for and being a “worrier” about health were two additional barriers identified by our study. Lack of preventative measures was cited as a barrier by our interviewees, which echoes the calls of health professionals to only offer genetic testing for genes with well-established clinical utility [49]. A UK study investigating population-based *BRCA* testing in the AJ population found issues around confidentiality, insurance, emotional impact, inability to prevent cancer, marriage ability and ethnic focus or stigmatisation were significantly associated with lower odds of uptake of *BRCA* testing and discriminated between accepters and decliners [38]. Having children resulted in stronger positive attitudes towards *BRCA* testing [38]. These findings are largely similar and complement our qualitative findings for women in the general population. Overall, we found high acceptability for unselected PGT in low-risk women, as it was also reported in the AJ population [38].

Two interviewees were Southeast Asian women, both who viewed PGT/OC risk stratification positively. Although this supports some findings of focus group interviews on hypothetical OC risk assessment by Hann, our sample size was too small to properly stratify and draw firm conclusions for ethnic minority outcomes [51]. Unlike Hann, no sociocultural nuances or barriers (marriageability, marital problems, issues of body privacy and shyness towards body exposure) were elicited during our interviews. Although the age of our group was similar (mean age = 53.7), there were some demographic differences between the studies with our interviewees being in full-time employment and university educated.

## 5. Conclusions

Unselected PGT is a novel approach that can overcome the limitations of the current clinical criteria/FH-based approach to genetic testing. It identifies all cancer susceptibility gene carriers, the majority of which are missed by FH-based testing. The combination of genetic (including polygenic risk scores) with non-genetic/epidemiological data improves the precision of risk estimates and enables population stratification using a personalised absolute OC risk estimate to increase identification of higher risk women, thereby maximising precision prevention. Our data show that attitudes towards unselected PGT/OC risk stratification were generally positive with high levels of satisfaction and reduced anxiety post-delivery of low-risk results. These outcomes are largely similar to those found in Jewish population testing studies. Most of the elicited facilitators/barriers were similar to those previously reported from testing high-risk families or with unselected *BRCA* testing in the AJ population. Communication of low-risk results must avoid over-reassurance and potential undermining of appropriate future help-seeking behaviour for symptoms, as low-risk does not equate to “no risk”. Larger implementation studies are needed to demonstrate satisfaction in those receiving intermediate/high-risk results as well as to assess long-term outcomes and cost effectiveness.

## Figures and Tables

**Table 1 diagnostics-12-01028-t001:** Participant characteristics.

ID	Age (Years)	Ethnicity	Marital Status	Employed	Parity	Lifetime Risk of OC (%)	Number of Relatives with OC	Number of Relatives with Non-Ovarian Cancers ***	Previous Genetic Test	Time from Results to Interview (Days)
01	47	Caucasian	Married	No	3	2.7	1 FDR	4	No	117
02	44	Southeast Asian	Cohabiting	Yes	1	0.7	0	0	No	117
03	60	Jewish	Married	Yes	2	1.2	0	1	No	120
04	57	Caucasian	Married	Yes	2	1.5	1 FDR	1	No	113
05	51	Caucasian	Divorced	Yes	2	0.6	0	0	No	110
06	69	Caucasian	Married	No	2	1.2	0	4	No	110
07	37	Caucasian	Married	No	3	1.0	0	2	No	116
08	33	Southeast Asian	Single	Yes	0	1.9	0	1	No	115
09	85	Caucasian	Widowed	No	0	0.6 *	0	4	Yes **	95

OC—ovarian cancer; FDR—first degree relative; SDR—second degree relative. No volunteers fulfilled the standard NHS clinical genetic testing criteria (based on 10% BRCA probability). * Ten-year ovarian cancer age-specific population risk derived from Cancer Research UK data (https://www.cancerresearchuk.org/health-professional/cancer-statistics/statistics-by-cancer-type/ovarian-cancer/incidence#heading-One) (accessed on 6 February 2022). We were unable to provide an individualised lifetime ovarian cancer risk, as the upper age limit of the risk prediction algorithm model is 80 years. ** JAK2 (Janus kinase 2)/CALR (Calreticulin) negative. *** Family history of non-ovarian cancers as follows: ID-01: 2 SDR leukaemia, 2 SDR bowel cancer; ID-03: 1 SDR bowel cancer; ID-04: 1 FDR gastric cancer; ID-06: 2 SDR melanoma, 2 SDR endometrial cancer; ID-07: 2 FDR leukaemia; ID-08: 1 FDR bowel cancer; ID-09: 1 FDR gastric cancer, 1 SDR melanoma and 2 SDR breast cancer.

**Table 2 diagnostics-12-01028-t002:** Facilitators and barriers to uptake of ovarian cancer genetic testing and risk prediction.

Facilitators	Quotes	Explanation
Social		
Altruism	“If I can help other people, I like to do that” (ID-06)	Individuals stated that altruism to help benefit other women in the future was a major motivator.
Involvement in ovarian cancer research studies	“Having been through the UKCTOCS [30], I thought well I might as well go ahead with it” (ID-06)	Participants stated that because they had been part of another OC research study (i.e., UKCTOCS), it had motivated them to undergo PGT/risk stratification, as taking part in research studies for the benefit of the wider community had been a “lifelong passion”.
Media publicity of cancers	“You hear now always the information coming out saying that one in three of us is going to have cancer at some point in life, but knowing that it might hit you at some point, so generally we’re thinking about the cancer, it’s out there all the time” (ID-07)	Widespread publicity from media campaigns on cancer-related issues positively influenced decision making as well as the media attention associated with Angelina Jolie undergoing *BRCA* testing.
Encouragement from relatives and friends	“People do seem really to want to do it, well within my circle, people say without question, of course I would do it” (ID-06)	Encouragement from family/friends motivated individuals to undergo testing.
Knowing someone who has had a genetic test	“I had had a conversation with her and she definitely recommended me to have the gene testing done” (ID-08)	Knowing someone who had undergone genetic testing and who had a positive experience influenced the decision of some participants.
**Demographic**		
Ethnicity	“Also I am Jewish” (ID-03)	Being Jewish was a strong motivator to undergoing testing/risk stratification, as individuals were aware that they were more likely to be carriers for certain genetic diseases.
Having children	“Mainly because of my sister and myself having two girls” (ID-03)	Individuals stated having children as a facilitator. This was rooted in the notion that if they were found to be at increased risk, then it could also affect their children as they could have inherited a genetic mutation. Therefore undergoing PGT/risk stratification themselves was beneficial to their children by proxy.
Family history of ovarian cancer	“20 years ago my mother had ovarian cancer when she died and it was really, really bad and at the time, I asked if there was any testing and they said that there wasn’t and that I was at a high risk of having it, but that there was nothing that they could do at that time” (ID-01)	Interviewees felt that having an FH of OC was a motivator due to the distress they experienced in watching their relatives dying from OC.
Family history of other cancers	“My two cousins died of breast cancer so I was quite interested” (ID-09)	Interviewees stated that an FH of other cancers was also a motivator, as it heightened awareness of cancer in general.
**Psychological**		
Curiosity	“I was very interested actually because I did wonder if I had one of these cancer genes” (ID-09)	Individuals stated curiosity as an important reason, which was linked to the desire to stay healthy.
Desire to stay healthy	“I’d rather be aware of what I’m predisposed to than not. Not that I would let it interfere with my day-to-day life but it’s more the fact that if I knew I had a predisposition then it would make me consider my lifestyle choices and change them to a beneficial way that although might not necessarily prevent it completely but it might improve my chances for it not to materialise” (ID-02)	Individuals expressed the need to learn as much as possible about their inherent genetic predispositions so that they had the opportunity to make better informed lifestyle choices to empower and give them greater control over their own health.
Future feelings of regret if developed ovarian cancer and passed up the opportunity for genetic testing/risk assessment	“At the time, I sort of twisted it round, I did have a few should I shouldn’t I moments but I flipped it over, how would I feel in x number of years if something happened and I could have done something about it? So in the end, I felt better to proceed” (ID-05)	Individuals felt that if they were to forgo this opportunity and subsequently developed OC, they may regret not being tested.
Ovarian cancer worry	“It was something that was always in the back of my mind” (ID-07)	Some cited OC worry as a facilitator, as they were seeking reassurance.
Unknown family history	“I have no idea why people in my family have passed away, it was really important to see if there was a risk because you just hear that somebody’s passed away and you have no idea what they’ve passed away of, so I couldn’t even refer or anything to that family medical history” (ID-02)	For some individuals, not knowing their family’s medical history and causes of death, were strong motivators. This was because they saw undergoing PGT as a way of providing themselves with insights into the potential causes of death for their relatives which could benefit them and their children.
**Logistical**		
Existing preventative measures	“I thought great, this must be a good thing, surely it’s better to know particularly with something like ovarian cancer, where you can do something about it, [and if I was found to be] high risk, I’d have been on my hands and knees begging you, to get them ovaries out” (ID-03)	The presence of established OC preventative measures to reduce risk encouraged individuals to undergo testing.
Difficulty in the early detection of ovarian cancer	“I know about how it creeps up unaware and isn’t clear or obvious until quite often it’s too late” (ID-01)	The difficulty in detecting OC early, poor OC prognosis and the ease of undergoing PGT were motivators.
Simplicity and ease of testing	“Well, it was a very simple test, a non-invasive test, well a blood test if you call that invasive but not really. But, yeah, it was a very simple test and I thought it all sounded pretty straightforward” (ID-08)	
**Barriers**	**Quotes**	**Explanation**
**Social**		
Stigma	“Any information where it can be used for positive gains, it can also be used for negative gains. So, I know there’s been some controversy about having genetic tests done, in case it affects insurance premiums etc. and again, being discriminated against in future employment, in case it’s something you’re asked to readily provide” (ID-04)	Interviewees felt that PGT had a stigma attached that was rooted in the notion that genetic results may be used against individuals by future employers and insurance companies.
Change in family dynamics	“The thought did occur to me because there was a possibility that the results might come out saying that I have a high chance of developing ovarian cancer and I guess that would have had implications in the family, but I didn’t want to dwell on it too much without knowing if this was the case. But if it was the case, I knew there was a possibility where I would have to think ethically, do I need to inform other family members?” (ID-07)	Some participants felt that finding a mutation may alter family dynamics, and it could raise ethical dilemmas as to whether to inform relatives.
**Psychological**		
Being a worrier	“I think it depends on how you see things really and how much of a worrier you are about your health” (ID-01)	Some cited “being a worrier” as a reason for not undergoing testing, as such individuals may not be able to cope mentally/emotionally with being told they had increased OC risk.
**Logistical**		
Insurance implications	“I think the only thing that you might consider and be concerned about is if you did have a problem and you were wanting to get insurance or something, then would you have to declare [your results]” (ID-03)	Implications on insurance may serve as a barrier to undergoing testing, especially if in the future there was an obligation to divulge results to insurance companies.
No known prevention	“If I was to learn you’ve got something where you can basically drop down dead, I wouldn’t want to know. I would only want to know if there was something you could do to reduce my risk” (ID-03)	Some cited no known preventative measures as a barrier.

UKCTOCS—UK Collaborative Trial of Ovarian Cancer Screening.

## Data Availability

The data presented in this study are available upon request from the corresponding author. The data are not publicly available due to the fact of confidentiality.

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
