# Peer review of "Unselected Population Genetic Testing for Personalised Ovarian Cancer Risk Prediction: A Qualitative Study Using Semi-Structured Interviews"

_diagnostics, 2022, doi:10.3390/diagnostics12051028_

Round 1
Reviewer 1 Report
I thank the academic editor for giving me the opportunity to re-evaluate this new version of the manuscript of Gaba F. et al. I believe, as already expressed the first time, that, although it is not a quantitative manuscript and with the need for comparative data, it is nevertheless a good example of the administration of a semi-structured interview to patients genetically at risk for ovarian cancer. Therefore I do not find any point of prejudice in this paper.
Author Response
We thank the reviewer for his/her comment.
Reviewer 2 Report
Gaba and colleagues proposed an interesting strategy to evaluate the risk of ovarian cancer in individuals. Through the development of a robust questionnaire, they proposed an innovative method for the real assessment of OC-risk. Overall, the manuscript is interesting, however, there are some issues that the authors have to address before publication:
1) As a multidisciplinary evaluation of ovarian cancer risk was performed, in the Introduction section, the authors have to describe how the multidisciplinary management of ovarian cancer has improved the diagnosis and prognosis of ovarian cancer patients. For this purpose, please see:
- PMID: 34132354
- PMID: 29770626
- PMID: 21055797
2) Please consider to move the following paragraph to the results section: “ Among the PROMISE-FS cohort, 102/103 participants received low-risk results, 0/103 intermediate-risk, and 1/103 high-risk, and it was therefore decided to focus recruitment on those with low-risk results, to enable information saturation.”;
3) Consider to use subheadings for the methods adopted;
4) In the Discussion section, please describe if other questionnaires are already available for the assessment of OC-risk in individuals with predisposing conditions;
5) In the conclusive remarks, please emphasize the novelty of the approach here adopted disclosing potential conflicts with other existing methods.
Author Response
RESPONSE TO REVIEWER 2
Gaba and colleagues proposed an interesting strategy to evaluate the risk of ovarian cancer in individuals. Through the development of a robust questionnaire, they proposed an innovative method for the real assessment of OC-risk.
Overall, the manuscript is interesting, however, there are some issues that the authors have to address before publication:
1) As a multidisciplinary evaluation of ovarian cancer risk was performed, in the Introduction section, the authors have to describe how the multidisciplinary management of ovarian cancer has improved the diagnosis and prognosis of ovarian cancer patients. For this purpose, please see:
- PMID: 34132354
- PMID: 29770626
- PMID: 21055797
Response
We have now added these remarks along with the suggested references to the first paragraph of the introduction.
Lines 58-61:
“In recent years, multi-disciplinary management of ovarian cancer (OC) has led to improved care and cancer outcomes for patients (1-3). Nevertheless, OC causes significant mortality due to its aggressive nature, late stage at presentation, and lack of an effective screening programme for general population women.(4) Therefore effective targeted preventive strategies are necessary and their validation and implementation requires similar collaboration between disciplines.”
2) Please consider to move the following paragraph to the results section: “ Among the PROMISE-FS cohort, 102/103 participants received low-risk results, 0/103 intermediate-risk, and 1/103 high-risk, and it was therefore decided to focus recruitment on those with low-risk results, to enable information saturation.”;
Response
Thank you for this suggestion, we have now moved this sentence to the results section.
Lines 182-184:
“Among the PROMISE-FS cohort, 102/103 participants received low-risk results, 0/103 intermediate-risk, and 1/103 high-risk, and it was therefore decided to focus recruitment on those with low-risk results, to enable information saturation.”
3) Consider to use subheadings for the methods adopted;
Response
Thank you for this suggestion to improve readability, we have now adopted 3 subheadings for the methods:
Line 134: “2.1. PROMISE-feasibility study design”
Line 141: “2.2. Qualitative study design”
Line 177: “2.3 Data analysis”
Point 4) In the Discussion section, please describe if other questionnaires are already available for the assessment of OC-risk in individuals with predisposing conditions;
Response
We thank the reviewer for this suggestion. We have added a paragraph detailing other risk-prediction models described in the literature and how they differ from this one.
Lines 395-404:
“ A few OC-risk prediction models have been published previously. However, these have been restricted to largely epidemiological/family-history based factors.(5, 6) Pearce et al incorporated a SNP based genetic score along-with epidemiological/hormonal/family-history based assessment but this model did not include moderate-high penetrance cancer susceptibility genes.(7) The risk-prediction algorithm used in our study is more comprehensive as it incorporates both genetic factors in terms of established cancer susceptibility genes as well common genetic variants (polygenic risk score) along-with non-genetic (epidemiological/family-history/hormonal/reproductive) factors in general-population women.(8) This has been validated in a prospective dataset from the UKCTOCS general population screening trial.(4, 8)
5) In the conclusive remarks, please emphasize the novelty of the approach here adopted disclosing potential conflicts with other existing methods.”
Response
Thank you for this comment. We have edited the conclusion to reflect this.
Lines 501-507:
“Unselected-PGT is a novel approach which can overcome the limitations of the current clinical-criteria/FH-based approach to genetic testing. It identifies all cancer susceptibility gene carriers, the majority of which are missed by FH-based testing. The combination of genetic (including polygenic-risk-scores) with non-genetic/epidemiological data improves the precision of risk-estimates and enables population stratification using a personalised absolute OC-risk estimate to increase identification of higher risk women thereby maximising precision prevention.”
“Unselected-PGT is a novel approach which can overcome the limitations of the current clinical-criteria/FH-based approach.”
Lines 518-522:
“Larger implementation studies are needed to demonstrate satisfaction in those receiving intermediate/high-risk results, and assessing long-term outcomes and cost-effectiveness.”
References
- Burton H, Chowdhury S, Dent T, Hall A, Pashayan N, Pharoah P. Public health implications from COGS and potential for risk stratification and screening. Nat Genet. 2013;45(4):349-51.
- Falzone L, Scandurra G, Lombardo V, Gattuso G, Lavoro A, Distefano AB, et al. A multidisciplinary approach remains the best strategy to improve and strengthen the management of ovarian cancer (Review). Int J Oncol. 2021;59(1).
- Suh DH, Chang SJ, Song T, Lee S, Kang WD, Lee SJ, et al. Practice guidelines for management of ovarian cancer in Korea: a Korean Society of Gynecologic Oncology Consensus Statement. J Gynecol Oncol. 2018;29(4):e56.
- Menon U, Gentry-Maharaj A, Burnell M, Singh N, Ryan A, Karpinskyj C, et al. Ovarian cancer population screening and mortality after long-term follow-up in the UK Collaborative Trial of Ovarian Cancer Screening (UKCTOCS): a randomised controlled trial. Lancet. 2021;397(10290):2182-93.
- Li K, Husing A, Fortner RT, Tjonneland A, Hansen L, Dossus L, et al. An epidemiologic risk prediction model for ovarian cancer in Europe: the EPIC study. Br J Cancer. 2015;112(7):1257-65.
- Pfeiffer RM, Park Y, Kreimer AR, Lacey JV, Jr., Pee D, Greenlee RT, et al. Risk prediction for breast, endometrial, and ovarian cancer in white women aged 50 y or older: derivation and validation from population-based cohort studies. PLoS Med. 2013;10(7):e1001492.
- Pearce CL, Stram DO, Ness RB, Stram DA, Roman LD, Templeman C, et al. Population distribution of lifetime risk of ovarian cancer in the United States. Cancer Epidemiol Biomarkers Prev. 2015;24(4):671-6.
- Lee A, Yang X, Tyrer J, Gentry-Maharaj A, Ryan A, Mavaddat N, et al. Comprehensive epithelial tubo-ovarian cancer risk prediction model incorporating genetic and epidemiological risk factors. J Med Genet. 2021.
Round 2
Reviewer 2 Report
The authors well addressed all of my comments. The manuscript can be accepted for publication after the editorial check.